

# Interaction of Val66Met BDNF and 5-HTTLPR polymorphisms with prevalence of post-earthquake 27-F PTSD in Chilean population

Juan-Luis Castillo-Navarrete[1,2,3], Benjamin Vicente[1,4], Kristin Schmidt[1,3,4], Esteban Moraga-Escobar[1], Romina Rojas-Ponce[1,3,5], Paola Lagos[1,5], Ximena Macaya[1,6] and Alejandra Guzman-Castillo[1,3,7]

[1] Programa Neurociencias, Psiquiatría y Salud Mental, NEPSAM, Universidad de Concepción, Concepción, Chile
[2] Departamento de Tecnología Médica, Facultad de Medicina, Universidad de Concepción, Concepción, Chile
[3] Programa Doctorado Salud Mental, Departamento de Psiquiatría y Salud Mental, Facultad de Medicina, Universidad de Concepción, Concepción, Chile
[4] Departamento de Psiquiatría y Salud Mental, Facultad de Medicina, Universidad de Concepción, Concepción, Chile
[5] Departamento de Farmacología, Facultad de Ciencias Biológicas, Universidad de Concepción, Concepción, Chile
[6] Facultad de Odontología, Universidad de Concepción, Concepción, Chile
[7] Departamento de Ciencias Básicas y Morfología, Facultad de Medicina, Universidad Católica de la Santísima Concepción, Concepción, Chile

Corresponding author
Alejandra Guzman-Castillo, aleguzman@ucsc.cl

## ABSTRACT

Post-traumatic stress (PTSD) disorder is a mental health condition that can occur after experiencing or witnessing a traumatic event. The 27-F earthquake that struck Chile in 2010 was one such event that had a significant impact on the mental health of the population. A study was conducted to investigate the prevalence of PTSD and its associated factors among survivors of this earthquake. The study was a longitudinal design, involving a sample of 913 patients aged 18 to 75 years who attended 10 Primary Care Centers in Concepción, Chile. The Composite International Diagnostic Interview (CIDI) was used to assess both depressive episodes (DE) and PTSD before and after the earthquake. The study also involved genotyping studies using saliva samples from the participants, specifically focusing on the Val66Met and 5-HTTLPR polymorphisms. Statistical analysis was performed to examine the association between different variables and the presence of PTSD. These variables included demographic factors, family history of psychiatric disorders, DE, childhood maltreatment experiences, and critical traumatic events related to the earthquake. The results showed that the incidence of post-earthquake PTSD was 11.06%. No significant differences were found between the groups of participants who developed post-earthquake PTSD regarding the Val66Met or 5-HTTLPR polymorphisms. However, a significant association was found between the concomitant diagnosis of DE and the development of post-earthquake PTSD. The presence of DE doubled the risk of developing post-earthquake PTSD. The number of traumatic events experienced also had a statistically significant association with an increased risk of developing post-earthquake PTSD. The study's limitations include the potential interference of different DE subtypes, the complexity of quantifying the degree of earthquake exposure experienced by each individual, and events entailing

social disruption, such as looting, that can profoundly influence distress. In conclusion, the study found that PTSD following the 27-F earthquake in Chile was associated with a concomitant diagnosis of DE and the number of traumatic events experienced. The study did not find a significant association between PTSD and the Val66Met or 5-HTTLPR polymorphisms. The researchers recommend that mental health professionals should prioritize the detection and treatment of concomitant depressive episodes and exposure to critical traumatic events in survivors of disasters. They also suggest that further research is needed to better understand the relationship between genetic factors and post-disaster PTSD.

# INTRODUCTION

Chile, given its geographical location, is a territory that is under constant threat from natural disasters (*Fernandez et al., 2017*). In fact, of the 10 most intense earthquakes in world history, two have occurred in this country. In 1960, in Valdivia city (9.5 Richter) and recently, on February 27, 2010 (27-F) (8.8 Richter) in central Chile (*Fernandez et al., 2020*; *Santos, Bymes & Lane, 2010*). The 27-F earthquake typifies many modern multifaceted natural disasters.

On the morning of February 27th, an earthquake occurred. It acted as the primary precipitating event for two subsequent disasters that occurred in rapid succession. A devastating tsunami affected and destroyed approximately 450 kilometres (*Leiva-Bianchi, Baher & Poblete, 2012*). This was followed by subsequent flooding, which occurred without proper warning. This was compounded by several days of looting and cuts in basic services in the epicentre region (*Garfin et al., 2014*; *Ramirez & Aliaga Sandoval, 2012*). As a result, this earthquake caused 500+ fatalities, and 12,000 injuries, and displaced 800,000+. Additionally, thousands of buildings were damaged or destroyed (*Santos, Bymes & Lane, 2010*).

Major traumatic events play a key role in the development of post-traumatic stress disorder (PTSD) (*Castro-Vale & Carvalho, 2020*; *Hori et al., 2020*; *Ortega-Rojas et al., 2017*). Major traumatic events include natural disasters, serious accidents, and war, among others (*Mojtabavi et al., 2020*; *Monson & Shnaider, 2014*; *Notaras & Van den Buuse, 2020a*). Alternatively, PTSD is a chronic course disorder that involves severe functional impairment. This is linked to an increase in reported physical illnesses, emergency visits, and hospitalizations/surgeries (*Quinones et al., 2020*; *Tuerk et al., 2013*). In this sense, its spectrum clinically includes re-experiencing the traumatic event, even in a safe context. PTSD is characterized by intense and persistent fear reactions and negative cognitive and mood alterations (*Hori et al., 2020*; *Mojtabavi et al., 2020*; *Quinones et al., 2020*). Therefore, individuals with this disorder have an excessive consolidation of memories associated with fear and its emotions (*Mojtabavi et al., 2020*; *Takei et al., 2011*).

Adverse experiences or traumatic stressors in childhood, adolescence, or adulthood have been widely linked to PTSD (*Kessler et al., 2010*; *Pereira et al., 2022*; *Wang, Shelton & Dwivedi, 2018*). Child maltreatment is harm or risk of harm to a child by a caregiver's act or omission. Also includes acts of physical, emotional, and sexual abuse and/or neglect (*World Health Organization (WHO), 2014*). Child maltreatment is among the strongest predictors of PTSD (*Dorrington et al., 2019*; *McLaughlin et al., 2017*). It affects up to 37.5% of children exposed to maltreatment (*Alisic et al., 2014*; *Scott, Smith & Ellis, 2010*). Other factors, such as previous trauma, gender, depressive episode (DE), and hereditary factors, have been linked to post-disaster PTSD (*Carr et al., 2013*; *Gallo et al., 2018*; *Hughes et al., 2017*; *Kessler et al., 2010*; *Li et al., 2021a*; *Pereira et al., 2022*). Individuals in high-risk disaster settings are heavily exposed, increasing the risk of developing PTSD and DE (*Fernandez et al., 2020*; *Norris et al., 2006*). Depressive psychopathology comorbid with PTSD is associated with similar neuropsychological, cognitive, and emotional regulation alterations (*Galatzer-Levy et al., 2013*; *Kachadourian, Pilver & Potenza, 2014*).

In neurobiological terms, PTSD's pathophysiology, progression, and maintenance involve multiple factors, presenting many questions (*Aksu et al., 2018*). PTSD could be a multi-dimensional disorder that consists of several subtypes with diverse neurobiological foundations (*De Berardis et al., 2015*; *De Berardis et al., 2019*). Various genetic factors influence stress reactions, with PTSD heritability up to 49% in some populations (*Li et al., 2021b*; *Wolf et al., 2018a*). BDNF and 5-HTTLPR genes are among the proposed PTSD vulnerability genes candidates (*Li et al., 2021a*; *Notaras & Van den Buuse, 2019*; *Notaras & Van den Buuse, 2020b*; *Zhang et al., 2017*). BDNF is involved in the maintenance of neuronal development, differentiation, and plasticity. Also is essential for maintaining brain physiological processes influencing, both memory and learning, appetite and sleep (*Karege et al., 2002*; *Lommatzsch et al., 2005*; *Nagahara & Tuszynski, 2011*).

A single nucleotide polymorphism (SNP) called Val66Met (rs6265, G/A) exists in the BDNF gene on 11p13. It lead to altered BDNF packaging and reduced release-dependent activity (*Hing, Sathyaputri & Potash, 2018*; *Nagahara & Tuszynski, 2011*; *Notaras & Van den Buuse, 2020b*). Val66Met is associated with cognitive changes, including memory impairment and reduced hippocampal activity (*Molendijk et al., 2011*; *Molendijk et al., 2012*; *Notaras & Van den Buuse, 2020a*). Chronic stress may potentiate fear circuitry in individuals carrying the Met variant. Thus, making them more susceptible to developing anxiety and fear-related disorders, including PTSD (*Hori et al., 2020*; *Notaras, Hill & Van Den Buuse, 2015*).

Traumatic events have been described to increase serotonin release in some brain regions (*Li et al., 2021a*; *Madsen et al., 2016*; *Xie et al., 2009*). The 5-HTT gene (SLC6A4) contains a polymorphic region that modifies the expression of the serotonin transporter (*Caspi et al., 2003*; *Li et al., 2021a*; *Rojas et al., 2015*; *Zhang et al., 2017*). The presence of a short (S) allele is associated with lower levels of the serotonin transporter. These levels are also affected by another polymorphism, A/G (rs25531), also known as the LG allele. The S allele of the 5-HTTLPR has a similar 5-HTT expression to the L and LG alleles. Less efficient 5-HTTLPR regulation and serotonin levels in LG and S allele carriers increase PTSD risk (*Madsen et al., 2016*; *Navarro-Mateu et al., 2019*; *Wolf et al., 2018b*; *Xie et al., 2009*).

Consequently, the presence of these alleles would increase the risk of developing stress-related disorders, including PTSD (*Xie et al., 2009*). This study aims to determine if BDNF and 5-HTTLPR variants increase post-earthquake PTSD risk. It offers genetic and contextual information on the development of PTSD after a natural disaster.

## MATERIALS & METHODS

### Design

A longitudinal study of a sample of patients, aged 18 to 75 years, who attended 10 Primary Care Centres in Concepción, Chile.

### Participants

The cohort of 937 participants included in this study corresponds to the previously described cohort by *Rojas et al. (2015)*. In 2005, the PREDICT-FONDEF project enrolled 2832 patients for follow-up, of whom 87.1% ($n = 2466$) completed the 12-month follow-up (*King et al., 2008*; *Vicente et al., 2016*). In 2011, 1,602 subjects were contacted and provided saliva samples for genotyping studies. 379 subjects were excluded due to inadequate samples and 136 subjects for not experiencing the catastrophic event. The resulting final sample was 937 participants (Fig. 1).

### Instruments

The Spanish-language version 2.1 of the Composite International Diagnostic Interview (CIDI) was used in the study (*World Health Organization (WHO), 1997*). It assessed both DE and PTSD before and after the 2010 Chilean earthquake. The CIDI is a structured psychiatric diagnostic tool with good psychometric properties and is widely used (*Andrews & Peters, 1998*; *Kessler & Üstün, 2004*; *Robins et al., 1988*). Additionally, there are no restrictions on its use. The CIDI is conducted by lay interviewers without the use of outside sources of information or medical records (*World Health Organization (WHO), 1997*). The translated version utilized (the official Spanish translation from the World Health Organization) has been validated in Chile (*Vicente et al., 2006*). A modified version of the CIDI PTSD module (section F) was used to assess post-disaster PTSD. Furthermore, only those with disaster-related PTSD were included. The Depressive Disorders module was utilized to diagnose DE (the period that has passed since the 2010 disaster). The CIDI provided reliable and standardized assessments of DE and PTSD, ensuring accuracy and validity. Concerning the sociodemographic information and associated risk factors, we briefly describe how they were obtained.

During the PREDICT-FONDEF study, a comprehensive set of environmental risk factors for PTSD was collected (in individuals without intellectual disabilities). These risk factors were compiled using an inventory from the PREDICT-Europe Project and were also based on known risk factors from earlier literature (*King et al., 2006*). This includes valid and reliable self-administered measures. The set of risk factors encompasses demographic factors, family history of psychiatric disorders, DE, childhood maltreatment experiences, and critical traumatic events related to the earthquake. The latter category includes the death of a family member, physical injuries, and damage or loss of housing.

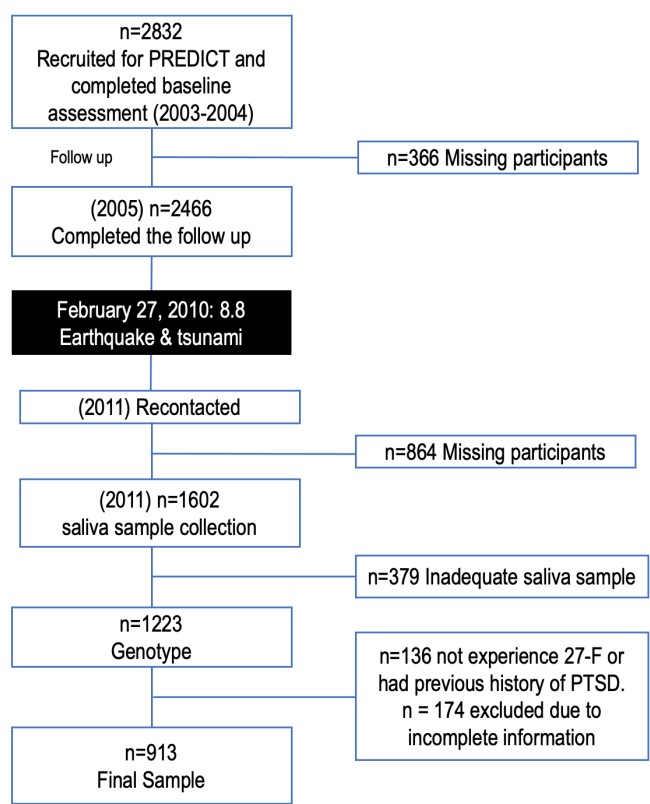

**Figure 1** **Flow diagram of excluded/ineligible individuals.** February 27, 2010: 8.8 earthquake & tsunami: In the early morning of February 27, the earthquake, the primary-precipitating event, was followed by two disasters in rapid succession. First, a devastating tsunami affecting and causing destruction over about 450 kilometers, and then the subsequent flooding that occurred without proper warning. PTSD: post-traumatic stress disorder.

## Ethical issues

According to the authors, all procedures used in this study and the 2008 revision of the 1975 Helsinki Declaration adhere to the ethical guidelines set forth by the pertinent national and institutional human experimentation committees. All procedures involving patients or human subjects received approval from the Ethics Committee of the Faculty of Medicine at the Universidad de Concepción. An informed consent form was signed by each person who consented to participate, which has been fully anonymized and cannot be identified through the manuscript (supplementary information).

## DNA extraction

Saliva samples were obtained, preserved, and transported using a DNA collection kit (Oragene-DNA G-500; DNA Genotek, Ottawa, Canada). DNA was then extracted using the salt precipitation method. The DNA concentration was then quantified using an Infinite® 200 PRO NanoQuant spectrophotometer (Tecan, Männedorf, Switzerland). Finally, DNA integrity was confirmed by agarose gel electrophoresis.

### Val66/Met BDNF genotyping

It was typed by restriction enzyme-based PCR (BsaA I). Specifically, the oligonucleotide partitions, sense F-1F (5′-ATCCCGGTGAAAGAAAGCCCTAAC-3′) and antisense F-1R (5′-CCCCTGCAGCCTTCTTCTTTTGTGTAA-3′) were used to amplify a PCR fragment 673 bp in length. The PCR fragments were then digested with the restriction enzyme BsaA I (New England Biolab, Ipswich, MA, USA). Specifically, this enzyme produces 3 fragments of 275, 321, and 77 bp when guanine is present at nucleotide 1249. In contrast, when cytosine is present at this position, 2 fragments of 321 and 352 bp are produced. Finally, the digested PCR products were separated on a 1.2% agarose gel.

### LPR genotyping

Genotyping of 5-HTTLPR for short and long alleles was performed by PCR (*Rojas et al., 2015*; *Wendland et al., 2006*). These alleles were amplified with the following partitions: sense F1 (5′-TCCTCCGCTTTGGCGCCTCTCTTCG-3′), and antisense R1 (5′-TGGGGGGTTGCAGGGGGGAGATCCTG-3′). These primers produce a 469 bp product for the short allele and a 512 bp product for the long allele. Then, the digestion of the PCR fragments was performed with the MspI I restriction enzyme (New England Biolab, Ipswich, MA, USA). As a result, the cut patterns SA: 469 bp, SG: 402 bp and 67 bp, LA: 512 bp and LG: 402 and 110 bp are obtained. Finally, these fragments were visualized on a 3% agarose gel. Additionally, all genotyping reactions were performed in duplicate.

### Met and 5-HTTLPR polymorphism analysis

Comparison groups were established to analyze the impact of these polymorphisms. Additionally, they were used to consider combinations of higher-risk *versus* lower-risk alleles for developing psychiatric disorders about each gene (*Bountress et al., 2017*; *Hori et al., 2021*). Thus, those homozygous alleles that would condition lower transcriptional and/or secretory activity are A/A and S/S' for Val66Met and 5-HTTLPR respectively. Consequently, the group at lower risk of developing psychiatric disorders are G/G and L/L for Val66Met and 5-HTTLPR respectively. Likewise, heterozygotes (G/A and L/S' for Val66Met and 5-HTTLPR respectively) were also compared with the higher-risk alleles, as they might also be at risk of developing psychiatric disorders.

### Data

The data used are available at https://doi.org/10.48665/udec/RQA125.

### Variables

To examine the interaction between various factors and the presence of PTSD, several variables were considered. Demographic confounding variables were obtained from the baseline CIDI assessment, while genetic variables included BDNF and 5-HTTLPR gene variants. A questionnaire created especially for the PREDICT study was used to collect sociodemographic and psychosocial data. This includes a family history of DE and experiences of childhood maltreatment (physical, emotional, and/or sexual). Consequently, the number of maltreatment forms was taken into account, irrespective of their type. Regarding the experience of the earthquake, a variable representing critical traumatic

events associated with the earthquake was included. This variable encompassed the death of a family member, being trapped under rubble, suffering serious or life-threatening physical injuries, and damage to or loss of housing.

## Statistical analysis

A significance level of $\alpha = 0.05$ was considered for all analyses. Specifically, RStudio version 2.15.2 (*R Core Team, 2023*) was used. Using the Kolmogorov–Smirnov and Shapiro–Wilk tests, 913 samples were tested for normality. Between-group differences in categorical variables for those with and without a PTSD diagnosis were calculated using the chi-square test, while differences in continuous variables were calculated using Student's $t$-test. In the regression analysis, independent associations between genetic predictors and PTSD risk were examined. A univariate logistic regression analysis with a logit link was used to determine odds ratios and 95% confidence intervals. To test the association between all variables (genetic, biological, and psychosocial) and PTSD risk, multivariate logistic regression analyses were performed. These models were built hierarchically based on theoretical reasoning. The first two models included only genetic risk factors, independently and with their interaction. The subsequent model incorporated additional biological variables, followed by a model including psychosocial factors. Finally, the last model encompassed the catastrophe model (Fig. 2).

## RESULTS

Table 1 displays the sociodemographic features of the 913 participants, with an 11.06% ($n = 101$) incidence of post-earthquake PTSD. Of the PTSD cases, 83.2% ($n = 84$) were female and 16.8% ($n = 17$) were male ($p = 0.375$). A total of 117 subjects (12.8%) experienced critical traumatic events associated with the earthquake while only 19 of those (18.8%) developed post-earthquake PTSD ($p = 0.079$). The mean number of traumatic events experienced was 2.2 ($+/-1.7$) for those who developed PTSD, compared to 1.2 ($+/-1.3$) for those who did not($p < 0.001$).

After analyzing the genetic data, no significant differences were found between the groups of participants who developed post-earthquake PTSD about the Val66Met or 5-HTTLPR polymorphisms or the combination of their alleles (A/A, G/A, and GG for Val66Met and L'/L', L'/S', and S'/S' for 5-HTTLPR) ($p = 0.419$ and $p = 0.344$, respectively). The study did not find any statistically significant differences when considering the number of forms of childhood maltreatment ($p = 0.459$), biological sex ($p = 0.375$), and level of schooling ($p = 0.590$).

To investigate the possible association between post-earthquake PTSD and various variables, a regression analysis was performed with increasing complexity, as depicted in Fig. 2 and supplementary information (Table S1). The univariate analysis for Val66Met and 5-HTTLPR considered the influence of the A allele (GA-AA) or S'/S' allele, respectively, on the development of post-earthquake PTSD, but no significant association was found ($p > 0.05$). The same was observed when examining the interaction between both alleles and the development of PTSD ($p = 0.949$), as well as for sex ($p = 0.313$) and age ($p = 0.345$).

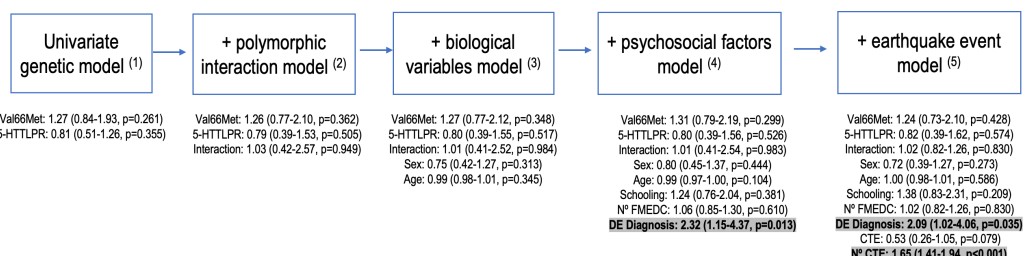

**Figure 2 Hierarchical construction of logistic regression models.** (1) Val66Met polymorphism (GG/GA-AA); 5-HTTLPR polymorphism (Other/ S'/S') (2)Interaction (Val66Met (GA-AA) - 5HTTLPR (S9/S9) Polymorphic Interaction) (3)Sex (Female / Male); Age (mean SD) (4) Schooling (Illiterate, basic, secondary or higher); N° FMEDC: Number of forms of maltreatment experienced during childhood (Mean SD; Variable created to group the number of forms of maltreatment experienced in childhood, independent of the type of maltreatment (physical, emotional or sexual)); DE Diagnosis: Diagnosis of a concomitant depressive episode (No/Yes) (5) CTE: Experience of a critical traumatic event associated with the earthquake (Death of a family member, Being trapped under rubble, Having physical injuries that required hospitalisation or were life-threatening, Severe damage to the dwelling or total los; No / Yes); N° CTE: Number of traumatic events experienced / witnessed (Mean SD).

When incorporating psychosocial variables, a statistically significant association was found between the concomitant diagnosis of DE and the development of post-earthquake PTSD ($p = 0.013$), but not for the number of forms of maltreatment experienced in childhood ($p = 0.610$). The presence of DE doubles the risk of developing post-earthquake PTSD (OR = 2.32, 95% CI [1.15–4.37], $p = 0.013$). When variables associated with the earthquake are added to the model, the significant association between the concomitant diagnosis of DE and the development of post-earthquake PTSD is maintained, doubling the risk of developing post-earthquake PTSD (OR = 2.09, 95% CI [1.02–4.06], $p = 0.035$).

It is important to note that the experience of a critical traumatic event associated with the earthquake was not statistically significant in the development of post-earthquake PTSD ($p = 0.079$). However, the number of traumatic events experienced had a statistically significant association with an increased risk of developing post-earthquake PTSD (OR = 1.65, 95% CI [1.41–1.94], $p < 0.001$). Thus, the number of traumatic events is a factor that increases the risk of developing post-earthquake PTSD by 1.65 times.

## DISCUSSION

The Chilean territory has a history of natural disasters, including earthquakes and volcanic eruptions, with significant impacts on the population. The 27-F earthquake provided a rare opportunity to examine the effects of a natural experiment on individuals who had previously undergone structured psychiatric diagnostic interviews. Our study involved implementing logistical procedures for adequate sampling, including PTSD screening and saliva sampling for genotypic studies. The recruitment of a remarkable number of participants was achieved, despite the lengthy reconstruction process in which the region was immersed.

After the earthquake, the prevalence of post-earthquake PTSD was found to be 11.06% (11.6% and 9.0% in women and men, respectively), with no significant difference from
**Table 1  Presence of polymorphisms, sociodemographic and psychosocial characteristics based on post-earthquake PTSD diagnosis.**

| Variables | Total (N = 913) | Post-earthquake PTSD | | p* |
|---|---|---|---|---|
| | | No (n = 812) | Yes (n = 101) | |
| Val66Met polymorphism | | | | |
| GG | 446 (48.8) | 402 (49.5) | 44 (43.6) | |
| GA | 421 (46.1) | 371 (45.7) | 50 (49.5) | 0.419 |
| AA | 46 (5.0) | 39 (4.8) | 7 (6.9) | |
| 5HTTLPR polymorphism | | | | |
| L'/L' | 151 (16.5) | 137 (16.9) | 14 (13.9) | |
| L'/S' | 444 (48.6) | 388 (47.8) | 56 (55.4) | 0.344 |
| S'/S' | 318 (34.8) | 287 (35.3) | 31 (30.7) | |
| Biological sex | | | | |
| Female | 724 (79.3) | 640 (78.8) | 84 (83.2) | 0.375 |
| Male | 189 (20.7) | 172 (21.2) | 17 (16.8) | |
| Age | | | | |
| Average (ds) | 55.1 (16.3) | 55.3 (16.6) | 53.5 (14.2) | 0.310 |
| Schooling | | | | |
| Illiterate | 94 (10.3) | 81 (10.0) | 13 (12.9) | |
| Basic | 275 (30.1) | 245 (30.2) | 30 (29.7) | 0.590 |
| Secondary | 410 (44.9) | 363 (44.7) | 47 (46.5) | |
| Higher | 134 (14.7) | 123 (15.1) | 11 (10.9) | |
| Experience of a critical traumatic event associated with earthquake[**] | | | | |
| No | 796 (87.2) | 714 (87.9) | 82 (81.2) | 0.079 |
| Yes | 117 (12.8) | 98 (12.1) | 19 (18.8) | |
| Numbers of traumatic events experienced/witnessed | | | | |
| Average (ds) | 1.3 (1.3) | 1.2 (1.3) | 2.2 (1.7) | **<0.001** |
| Number of forms of maltreatment experienced in childhood[***] | | | | |
| 0 | 481 (52.7) | 434 (53.4) | 47 (46.5) | |
| 1 | 188 (20.6) | 164 (20.2) | 24 (23.8) | 0.459 |
| 2 | 179 (19.6) | 155 (19.1) | 24 (23.8) | |
| 3 | 65 (7.1) | 59 (7.3) | 6 (5.9) | |

**Notes.**

[*]Chi-square for categorical variables, $t$-test for numerical variables.

[**]Death of a family member, having been trapped under rubble, having had physical injuries that required hospitalization or were life-threatening, Severe damage to the dwelling or total loss.

[***]Variable created to group the number of forms of maltreatment experienced in childhood, independent of the type of maltreatment (physical, emotional or sexual).

the literature's reported prevalence of PTSD (9 to 11%) for those who have been exposed to a traumatic event (*Abeldaño et al., 2014*; *Campos et al., 2022*). Before the earthquake, in Chile in 2009, a prevalence of 4.4% was identified in the population over 15 years of age, which was considered a reference value (*Benítez et al., 2009*).

When comparing prevalence rates of PTSD between studies, it is important to consider several factors such as the timing of the stressful event. For example, a study on the 27-F disaster conducted by the Chilean government 3 to 4 months after the earthquake reported a

national prevalence of 11.1%, with higher rates in the Province of Concepción (28.4%) and lower rates in regions unaffected by the earthquake (4.4%) (*Abeldaño et al., 2014*; *Larrañaga & Herrera, 2010*). Another study on adolescents in the city of Chillán, located inland from Concepción, found a PTSD prevalence of 20.4% at 7 months post-earthquake (*Díaz, Quintana & Vogel, 2012*). These variations in prevalence rates highlight the importance of considering context and timing when interpreting results.

There may be several factors that account for variations in the prevalence of post-earthquake PTSD reported in different studies, including differences in timing and the magnitude of exposure to the disaster. For instance, the Nepal earthquake of April 25, 2015, has been the subject of several studies, highlighting differences in prevalence rates depending on the timing and context of exposure. One study focusing on certain districts in the Kathmandu Valley found a prevalence of post-earthquake PTSD of 15.9% at 6 months (*Hatori & Bhandary, 2022*), while another study focusing on other districts in the same region reported a prevalence of 5.2% at 4 months (*Kane et al., 2018*). Differences in the socio-demographic composition of the samples and exposure context may explain some of these differences (*Hatori & Bhandary, 2022*).

In Chile, 73.6% of children reportedly experience physical or emotional violence from their parents or relatives (*UNICEF, 2000*). Nevertheless, our investigation did not reveal a significant association between the number of forms of childhood abuse experienced, irrespective of abuse type (physical, psychological, or sexual), and the development of post-earthquake PTSD. This lack of association might be attributable to the heterogeneity of traumatic exposures documented in the existing literature. Additionally, only a small number of individuals who developed post-earthquake PTSD reported experiencing maltreatment. Consequently, future research examining individuals with a history of childhood maltreatment who encounter a comparable disaster may provide further insights into this matter.

Numerous studies have identified a higher prevalence of PTSD among women (*Abeldaño et al., 2014*; *Hatori & Bhandary, 2022*; *Maya-Mondragón et al., 2019*); however, our investigation did not detect any sex-related differences. The increased vulnerability of women to PTSD has been attributed not only to biological factors but also to variations in socialization processes and formative childhood experiences (*Abeldaño et al., 2014*; *Breslau & Anthony, 2007*), in conjunction with exposure to trauma itself. As a result, the analogous exposure of men and women to the context of the 27-F earthquake might explain the absence of observed disparities. Furthermore, the loss of employment sources for men, who constituted the primary economic support for numerous affected households, should also be taken into account.

In the present study, we observed that a concomitant diagnosis of DE was significantly associated with an elevated risk of developing post-earthquake PTSD (OR = 2.32, 95% CI [1.15–4.37], $p = 0.013$). This risk was marginally reduced when accounting for the experience of the seismic event in the logistic regression model (OR = 2.09, 95% CI [1.02–4.06], $p = 0.035$). Notwithstanding, a concomitant DE diagnosis still doubled the risk of developing post-earthquake PTSD at 12 months. It is crucial to consider the potential that numerous individuals were not diagnosed with DE at the time of the 27-F, and that

the disaster merely exacerbated their symptoms, in conjunction with the concurrent development of post-earthquake PTSD.

It is imperative to emphasize that while DE can manifest as highly heterogeneous conditions, encompassing various subtypes such as anxious, melancholic, psychotic, or suicidal ideation, our study did not differentiate between these subtypes. This limitation arose from methodological constraints and the relatively small number of evaluated individuals who developed PTSD. A significant aspect to acknowledge is that the earthquake transpired during the early morning hours (03:38 AM), resulting in the earthquake and its ensuing events being experienced in a communal or familiar context, encompassing family gatherings, shared meals, and mutual support among neighbours. This scenario fosters coping strategies at both individual and collective levels. Our study's findings highlight that the experience of a single critical traumatic event associated with the earthquake did not constitute a significant variable. Conversely, witnessing more than one critical traumatic event linked to the earthquake emerged as a significant factor, escalating the risk of developing earthquake-induced PTSD by 1.65 times (OR = 1.65, 95% CI [1.41–1.94], $p < 0.001$).

Given the substantial disparity between trauma exposure and PTSD incidence, it is vital to enhance our understanding of genetic factors that may influence susceptibility to post-earthquake PTSD. Therefore, we aimed to evaluate whether the presence of BDNF and 5-HTTLPR genetic variants is associated with an increased risk of post-earthquake PTSD. Studies involving the Val66Met polymorphism have proposed that substitution with the Met allele leads to modified intracellular packaging and regulation of BDNF secretion, consequently decreasing brain BDNF levels. This deficiency in BDNF-induced intracellular signalling could adversely impact cortex-driven fear memory extinction (*Andero & Ressler, 2012*; *Young et al., 2021*) and heighten sensitivity to trauma exposure threat (*Ney et al., 2021*). This is in line with findings of a higher PTSD prevalence among carriers of at least one Met allele (*Notaras, Hill & Van Den Buuse, 2015*; *Pitts et al., 2020*). Furthermore, Met allele carriers exhibit decreased prefrontal cortex activity and increased amygdala activation without improved fear extinction, according to functional MRI studies in healthy individuals (*Lonsdorf et al., 2015*; *Ney et al., 2021*). The Met allele does not, however, appear to have a general impact on the symptoms of PTSD, according to several meta-analyses (*Bountress et al., 2017*; *Bruenig et al., 2016*; *Wang, 2015*), although a marginal effect has been described when comparing trauma-exposed subjects with and without PTSD (*Bruenig et al., 2016*). Furthermore, no significant findings have been reported in genome-wide association studies (GWAS) (*Bountress et al., 2017*; *Stein et al., 2016*).

In alignment with the aforementioned findings, our study did not identify a significant association between the Met allele and the incidence of post-earthquake PTSD, despite evidence suggesting that Met allele carriers may be more susceptible to developing anxiety- and fear-related disorders, including PTSD (*Hori et al., 2020*; *Notaras, Hill & Van Den Buuse, 2015*). As previously noted, DE can exhibit considerable heterogeneity. Therefore, in light of the existing literature, the Val66Met polymorphism may have distinct roles in these diverse subtypes (*Martinotti et al., 2016*; *Orsolini et al., 2020*). The presence of diverse

populations and numerous unidentified variables could account for this observation, necessitating further research to determine the potential existence of such associations.

While 5-HTTLPR has been thoroughly investigated about trauma, its association with PTSD presents mixed evidence (*Bountress et al., 2017*; *Valente et al., 2011*). It has been suggested that carriers of the LG and S alleles may be less effective in maintaining optimal levels of extracellular serotonin, thereby elevating the risk of developing stress-related disorders (*Li et al., 2021a*; *Madsen et al., 2016*; *Xie et al., 2009*). Specifically, some researchers have reported that individuals carrying at least one ''S'' allele are more susceptible to adverse environments (*Bountress et al., 2017*; *Navarro-Mateu et al., 2013*). However, our study did not identify a statistically significant association between the LG and S alleles and the incidence of post-earthquake PTSD. Furthermore, logistic regression analysis also failed to reveal a significant joint association between Val66Met and 5-HTTLPR concerning the incidence of post-earthquake PTSD.

Concerning limitations, we have already discussed the potential interference of different DE subtypes and the Val66Met polymorphism in our results. Another complex aspect to quantify is the degree of earthquake exposure experienced by each individual, which includes the scope of destruction and its influence on people's perceptions and ensuing distress. Moreover, events entailing social disruption, such as looting, can profoundly influence distress, leading to a disruption of the worldview that presupposes community safety and trust among neighbours (*Garfin et al., 2014*). The impact of such events on the development of PTSD cannot be dismissed.

Based on our study's findings, we suggest that clinical care for earthquake victims should be cognizant of the significant role that critical traumatic experiences and concomitant depressive episodes play in the development of post-disaster PTSD. From a clinical standpoint, incorporating assessments of depressive episodes and exposure to critical traumatic events when treating individuals affected by a catastrophe would be beneficial. Early detection and intervention in these aspects could help mitigate the risk of developing post-disaster PTSD. Additionally, our findings underscore the need for further research to explore the role of genetic factors in susceptibility to post-disaster PTSD. In summary, we recommend that mental health professionals be vigilant for depressive episodes and critical traumatic experiences in individuals who have experienced a catastrophe to provide early and effective interventions to prevent the onset of PTSD, and further research is needed to better understand the relationship between genetic factors and post-disaster PTSD.

The practical implications of our study suggest that mental health professionals should prioritize the detection and treatment of concomitant depressive episodes and the exposure to critical traumatic events in survivors of disasters, such as earthquakes. These factors significantly contribute to the development of post-disaster PTSD, and early interventions can potentially mitigate this risk. As for future research directions, our study indicates a need for a more comprehensive understanding of the genetic predispositions to post-disaster PTSD. Although our study did not find a significant association between the BDNF and 5-HTTLPR genetic variants, the role of genetic factors should not be discounted. Future researchers are encouraged to replicate our study in the context of different types of disasters and explore other genetic variants that could influence the development of PTSD. This

will not only validate our findings but also broaden the understanding of PTSD following disasters, potentially leading to more effective prevention and treatment strategies.

## CONCLUSIONS

In conclusion, this study illustrates that PTSD is a multifaceted phenomenon. According to the proposed final regression model, a concomitant diagnosis of depressive episodes doubles the risk of developing post-earthquake PTSD at 12 months. Furthermore, witnessing more than one critical traumatic event associated with the earthquake also poses a risk for the development of post-earthquake PTSD.

## ACKNOWLEDGEMENTS

(i) We would like to thank to Mr. Silverio Torres who played a crucial role in data filtering, data sheet and table preparation, and data analysis. His expertise and dedication greatly assisted us in ensuring the accuracy and reliability of the results. We are sincerely grateful for his efforts and are pleased to recognize his contribution. (ii) Also, we would like to thank to our study participants for their involvement. (iii) Some sections of this article were written with the help of the GPT-4 AI model. However, the results of this study are presented clearly, honestly, and without fabrication, falsification, or inappropriate data manipulation.

### Funding

This work was supported by Grant FONDECYT N°1110687 (Fondo Nacional de Desarrollo Científico y Tecnológico), Grant 218.087.043-1.0 of the Vice Rector of Research and Development (VRID) of the Universidad de Concepción, and a PhD scholarship in Chile ANID N°21201061. The funders had no role in study design, data collection and analysis, decision to publish, or preparation of the manuscript.

### Grant Disclosures

The following grant information was disclosed by the authors:
Fondo Nacional de Desarrollo Científico y Tecnológico: 218.087.043-1.0.
Vice Rector of Research and Development (VRID) of the Universidad de Concepción.

### Competing Interests

The authors declare there are no competing interests.

### Author Contributions

- Juan-Luis Castillo-Navarrete conceived and designed the experiments, analyzed the data, prepared figures and/or tables, authored or reviewed drafts of the article, and approved the final draft.
- Benjamin Vicente conceived and designed the experiments, authored or reviewed drafts of the article, and approved the final draft.

- Kristin Schmidt analyzed the data, authored or reviewed drafts of the article, and approved the final draft.
- Esteban Moraga-Escobar analyzed the data, prepared figures and/or tables, authored or reviewed drafts of the article, and approved the final draft.
- Romina Rojas-Ponce performed the experiments, authored or reviewed drafts of the article, and approved the final draft.
- Paola Lagos performed the experiments, authored or reviewed drafts of the article, preparation and conservation of genetic material, and approved the final draft.
- Ximena Macaya analyzed the data, authored or reviewed drafts of the article, prepare data base, and approved the final draft.
- Alejandra Guzman-Castillo conceived and designed the experiments, analyzed the data, prepared figures and/or tables, authored or reviewed drafts of the article, and approved the final draft.

## Human Ethics

The following information was supplied relating to ethical approvals (i.e., approving body and any reference numbers):

The Ethics Committee of the Faculty of Medicine of the Universidad de Concepción, Chile approved the study.

## Data Availability

Data is available at the Universidad de Concepción:

Juan Luis, Castillo Navarrete; Benjamín, Vicente; Kristin, Schmidt; Esteban, Moraga-Escobar; Romina, Rojas-Ponce; Paola, Lagos; Ximena, Macaya; Alejandra, Guzmán-Castillo, 2023, ''Interaction of Val66Met BDNF and 5-HTTLPR polymorphisms with prevalence of post-earthquake 27-F PTSD in Chilean population'', https://doi.org/10.48665/udec/RQA125, Repositorio de Datos - UdeC, V1, UNF:6:YCy7lRxnON4yy2kbmYKdpQ== [fileUNF].

## Supplemental Information

Supplemental information for this article can be found online at http://dx.doi.org/10.7717/peerj.15870#supplemental-information.

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
