# Peer review of "Interaction of Val66Met BDNF and 5-HTTLPR polymorphisms with prevalence of post-earthquake 27-F PTSD in Chilean population"

_PeerJ, doi:10.7717/peerj.15870_

## Round 0.1 · original submission · Major Revisions

I have now received the reviewers' comments on your manuscript. They have suggested some major revisions to your manuscript. Therefore, I invite you to respond to the reviewers' comments and revise your manuscript.

Reviewer 1 ·

Basic reporting

In the present study, the Authors aimed to evaluate whether the presence of genetic variants of BDNF and 5-HTTLPR might be associated with an increased risk of post-earthquake PTSD. In Authors opinion, that I agree, This is an opportunity to provide both genetic and contextual information related to a natural disaster and the subsequent development of PTSD.
Overall, I found this study timely, original, well-conducted and scientifically sound: I believe that it adds something new to the existing literature.
However, I have some suggestions aimed at improving the quality of the paper, and these are outlined below:
1) In the introduction, a brief note on the fact that PTSD might be a multidimensional disorder which includes several subtypes with different neurobiological underpinnings, should be added with appropriate references (see dois: 10.2174/1389450116666150506114108 and 10.1080/13651501.2019.1699575).
2) The Authors wrote that the initial sample consisted of 2,466 patients who completed the 12-month follow-up of the PREDICT-FONDEF Project in 2005. This corresponds to 87.1% of the total patients who began the follow-up stage (n=2,832), but the final sample consisted of 937 participants. I believe that isn’t clear whether the participants were consecutive or randomly selected. And how many subjects were screened, but not included for any reason (please specify why).
3) Was also the presence of an intellectual disability evaluated, how, and used as an exclusion criterion?
4) As well epressive episodes (DE) might be very heterogeneous, with different subtypes (i.e. anxious, melancholic, psychotic, with suicidal thoughts etc.) and this might have contributed to the study results. I suggest to add the lacking of evaluation of the depressive episodes subtypes in limitation. Besides BDNF might play a different role in different subtypes and this should be briefly discussed with appropriate references (please see dois 10.1093/ijnp/pyw003 and 10.30773/pi.2019.0171).
5) Translating into everyday “real world” clinical practice, what possible clinical shreds of evidence might arise from the present study on a very clinical point of view and what the Authors do recommend to improve practice?

Experimental design

Please, see above

Validity of the findings

Please, see above

Additional comments

Please, see above

·

Basic reporting

This manuscript by Castillo-Navarrette and coworkers titled ‘Interaction of Val66Met BDNF and 5-HTTLPR polymorphisms with prevalence of post-earthquake 27-F PTSD in Chilean population’ reports lack of association between Val66Met BDNF and 5-HTTLPR polymorphisms and post-traumatic stress disorder (PTSD) following February 27, 2010, earthquake. This study is interesting. I appreciate the authors for conducting a comprehensive statistical analysis.
I think the following suggestion will help in improving the manuscript.
1. Lines 59-67 in introduction appear difficult to comprehend and are largely vague. I feel these lines are not necessary to introduce the study. I suggest that this manuscript would read well without these lines. Please exclude this paragraph and start the introduction with the paragraph beginning from line 68 in the current version of the manuscript.
2. This manuscript requires proof reading by a fluent English speaker who understands the science presented.
3. I suggest authors to remove the following portion from lines 75-76 to avoid subjective judgements. ‘due to failure of the Chilean …………. (SHOA)’.
4. While the results obtained in the study are interesting, the way they were written is confusing. For example, there are phrases/clauses in the sentences that are not needed to be written. In line 267, ‘Considering the magnitude of 27-F’ can be removed. This sentence reads well if it is written like the following: A total of 117 subjects (12.8%) experienced critical traumatic event associated with the earthquake while only 19 of those (16.2%) developed post-earthquake PTSD (p=0.079). Similarly, please edit line 269-301. It is not possible to point out each such sentence in this review, but I listed a few lines that contain such unusual sentences. Lines 274, 276, 278, 281, and 297 etc.
5. Please include earthquake and its year in the flow chart.

Experimental design

No comments

Validity of the findings

No comments

Reviewer 3 ·

Basic reporting

This study investigated the association of BDNF and HTTLPR polymorphisms with PTSD occurred after 2010 earthquake in Chilean subjects who were followed up regularly as part of PREDICT study. This is well designed. Statistical analysis was appropriately conducted. Careful editing of the language is needed to improvise the presentation of data in this manuscript. I have the following concerns.
a) The English language should be improved to ensure that an international audience can clearly understand your text. I suggest you have a colleague who is proficient in English and familiar with the subject matter review your manuscript or contact a professional editing service. Improving the clarity in text will make this manuscript more interesting. Introduction and results section require significant improvement in language. The use of ‘when considering….’ while writing results seems odd.
b) The authors cited a paper for PREDICT assessment. I ask them to give more details on how the subjects were recruited and followed up. Also explain if there are any confounding factors that could influence the outcomes described in the study.
c) The authors may have put their best efforts to write this manuscript well. However, the text in some sections of the manuscript is unclear and there were some incomplete sentences. For example, 1) Line 60: What do ‘criterion’ and ‘validity of knowledge’? and 2) Line 65: ‘They have such characteristics. What are those characteristics. This sentence appears to be incomplete and incomprehensible.
d) Please expand PTSD in the abstract and text.

Experimental design

I have no comments

Validity of the findings

I have no comments

Reviewer 4 ·

Basic reporting

-

Experimental design

-

Validity of the findings

-

Additional comments

I would like to thank you for inviting me to review this manuscript. I read the article carefully. This longitudinal study aimed at investigating the interaction of Val66Met BDNF and 5-HTTLPR polymorphisms with prevalence of post-earthquake 27-F PTSD in Chilean population. Although the current article is scientifically valuable, the presence of some minor issues prevents it from being published at this stage of the review process.

I am a bit confused about the final sample of participants. In the participants subheading and Figure 1, the numbers presented about the final sample of participants seem uncertain. My suggestion is that the authors revise this section.  Also, in the statistical analysis subheading, 913 people were included in the analysis, but it is not clear why the remaining 24 people were excluded from the analysis.

It is suggested that in the materials and methods section, a paragraph about ethical considerations, compliance with confidentiality conditions and obtaining an informed consent form from the participants should be provided.

I think that explaining practical implications and future directions in the discussion section can help other researchers to replicate and expand the results of the present study.

Before using acronyms, define them separately in the abstract and main text.

Finally, I strongly recommend that the paper be checked by a fluent English speaker or professional English editing service before re-submission.

---

## Round 0.2 · Minor Revisions

Many thanks for submitting the revised manuscript in PeerJ journal. Based on one reviewer's opinion, this version of article needs some minor revisions. I invite you responding carefully to the reviewer's comments

Reviewer 1 ·

Basic reporting

The paper is improved and worthy of publication

Experimental design

The paper is improved and worthy of publication

Validity of the findings

The paper is improved and worthy of publication

Reviewer 4 ·

Basic reporting

-

Experimental design

-

Validity of the findings

-

Additional comments

Thank you for the update. However, there are still concerns that prevent me from accepting the revised paper:

The abstract is not acceptable in its current form. In the abstract, there is no mention of study time, study setting and sample size. Although PeerJ word limits specify a maximum of 500 words in the abstract section, however a typical abstract should only be about 250 to 300 words.

Page 8, lines 135-140: The explanation in this paragraph is not consistent with what is presented in Figure 1.

---

## Round 0.3 · accepted · Accept

Many thanks for addressing all the issues.